# New Perspectives on Avian Models for Studies of Basic Aging Processes

**DOI:** 10.3390/biomedicines9060649

**Published:** 2021-06-07

**Authors:** James M. Harper, Donna J. Holmes

**Affiliations:** 1Department of Biological Sciences, Sam Houston State University, Huntsville, TX 77341, USA; 2Department of Biological Sciences and WWAMI Medical Education Program, University of Idaho, Moscow, ID 83844, USA; djholmes@uidaho.edu

**Keywords:** animal models, avian aging, biogerontology, birds, erythrocytes, free radical hypothesis, geroscience, lifespan, oxidative damage, senescence

## Abstract

Avian models have the potential to elucidate basic cellular and molecular mechanisms underlying the slow aging rates and exceptional longevity typical of this group of vertebrates. To date, most studies of avian aging have focused on relatively few of the phenomena now thought to be intrinsic to the aging process, but primarily on responses to oxidative stress and telomere dynamics. But a variety of whole-animal and cell-based approaches to avian aging and stress resistance have been developed—especially the use of primary cell lines and isolated erythrocytes—which permit other processes to be investigated. In this review, we highlight newer studies using these approaches. We also discuss recent research on age-related changes in neural function in birds in the context of sensory changes relevant to homing and navigation, as well as the maintenance of song. More recently, with the advent of “-omic” methodologies, including whole-genome studies, new approaches have gained momentum for investigating the mechanistic basis of aging in birds. Overall, current research suggests that birds exhibit an enhanced resistance to the detrimental effects of oxidative damage and maintain higher than expected levels of cellular function as they age. There is also evidence that genetic signatures associated with cellular defenses, as well as metabolic and immune function, are enhanced in birds but data are still lacking relative to that available from more conventional model organisms. We are optimistic that continued development of avian models in geroscience, especially under controlled laboratory conditions, will provide novel insights into the exceptional longevity of this animal taxon.

## 1. Introduction: Avian Models in Geroscience

Researchers focused on the basic biology of aging and longevity in humans and model organisms currently use an intellectual schema that identifies “hallmarks” (or alternatively, “pillars”) of aging—key cellular, molecular, and metabolic processes that ultimately lead to the loss of homeostasis and resiliency to stress and disease that characterize organismal aging and, ultimately, death (for examples, see [1,2]). Versions of the schema change over time to accommodate evolving research perspectives. However, the processes implicated as hallmarks include, among others, effects of oxidative stress and damage (including damage to DNA, lipids and proteins); changes in telomere attrition rates; dysregulation of metabolic processes; and disruption of various cell-signaling pathways. All are thought to play a role in the progressive loss of physiological resilience to stress that characterizes organismal aging. A primary motivation of basic research in biogerontology or “geroscience” is to find ways to slow aging, often by identifying molecular or metabolic targets for intervention in aging-related degenerative processes.

Historically, vertebrate models for aging studies have been limited mainly to small, short-lived, inbred species (e.g., rodents) with an established body of information about their husbandry, physiology, cell biology, or genetics. (Other “traditional” model species include inbred fruit flies, the roundworm *C. elegans,* and brewer’s yeast *(Saccharomyces)*). Since the 1990s, however, there has been rising interest in developing a more diverse array of comparative approaches and animal models, including birds and other unusually long-lived vertebrates. New animal models that show promise include birds, since many bird species are longer-lived than mammals of similar body weight, despite the high metabolic rates characterizing class Aves overall (for reviews, see [3,4,5,6,7,8,9,10]). Within any taxonomic group there are intriguing cases of extreme longevity. For example, microchiropteran bats are exceptionally long-lived mammals for their size, in contrast to the Norway (i.e., laboratory) rat, which is much shorter-lived than mammals of similar body size. This is not to argue against the value of studies of laboratory rodents; rather, that long-lived exceptions to the norm can suggest useful “alternative” models for aging studies.

The need for more integrative avian aging studies parallels a lack of development of small, nonhuman primate models for geroscience, despite their potential advantages [11,12]. An obvious downside of using long-lived experimental animals is that funding agencies will generally not support projects using animal species that are known to live up to a decade or more. (On the other hand, there has been a continuous search over the past two decades for non-domesticated, *short-lived* vertebrate models of aging (e.g., annual killifishes in the genus *Nothobranchius* (especially *N. furzeri*)*,* with a maximum lifespan of one to two years [13,14]).

Nevertheless, there are bird species with lifespans amenable to a typical grant funding cycle. Wild house sparrows (*Passer domesticus*) can live at least 3–4 years in captivity (see, for example, [15]); this species has been used extensively for studies (including laboratory studies) of avian reproduction and physiology. Quantifiable differences in survival and fertility among groups of captive or domestic zebra finches (*Taeniopygia guttata;* maximum captive lifespan of 8+ years) can also be discerned within a reasonable time frame. Moreover, domestic Japanese quail (*Coturnix*) typically have a lifespan of 2–3 years, although cases of individuals living for more than five years in the lab have been reported [16]. Thus, the lifespanss of some birds are comparable to the typical lifespan of a laboratory mouse, and even shorter than that of most laboratory rat strains [17].

We have presented more detailed rationales for using avian and other “nontraditional” models in biological aging studies in a number of past reviews [3,4,5,6,7]. In this article, our aim is to highlight examples of recent—and promising—aging-related research using birds as model organisms. The examples we highlight here are varied, and include work both on wild, free-ranging birds as well as domesticated birds in captivity.

One of the challenges of using avian models for aging studies has been adapting putative molecular “biomarkers” used in more conventional aging research. Bird biologists interested in developing markers of stress or damage have tended to focus on two main areas: oxidative stress (usually as measured using blood) and telomere shortening. Butcomparative cell-culture techniques have also been developed to measure resilience of developing birds or avian cells to various stressors or toxins in vitro. Another fruitful line of work has involved in ovo hormonal manipulation of avian development and lifespan. We will touch briefly on each of these, and also mention some recent studies of dysregulation of energy metabolism, aging-related changes or instability at the whole-genome level, or from other “-omics” perspectives. Finally, we will highlight some intriguing recent studies focused on birds’ potential for neuroregeneration, and highlight some work investigating aging-related degeneration, damage, and dysregulation in key brain areas that control song learning in birds.

## 2. Avian Aging Studies in Field vs. Laboratory

Many studies that have included an aging component or a “life history” perspective have been conducted in the wild bird populations with known age structures and vital (reproductive and survival) rates. There are a number of published reports from long-term field studies using well established demographic approaches developed by avian population biologists and the bird conservation research community (for more thorough reviews, see [8,18,19]). Naturalistic studies of avian population biology always come with a set of potential confounds, including episodic variation in food availability, extremes of temperature or moisture, predation, or disappearance of individual birds for unknown reasons. When individuals of extreme ages are lost in such studies, mortality cannot clearly be attributed to aging or other intrinsic sources of mortality.

These potential confounds were investigated in a recent controlled study of captive zebra finches (*Taeniopygia guttata*), in which food availability and environmental conditions were systematically varied to ascertain their effect on survival [20]. Briefly, zebra finches subjected to a simulated “harsh” environment exhibited a significant increase in mortality rate when food availability was limited. Food availability, however, had no effect on mortality in birds maintained under “benign” conditions. This type of study serves as a caution against drawing firm conclusions about avian aging rates and mortality based on data from the field.

There are more examples of using zebra finches for studies of aging-related phenomena, showing that work with this species can be practical and productive [21]. For example, domesticated zebra finches have been shown to exhibit marked immunosenescence in both males and females [22], as well as a linear age-related decline in basal metabolic rate (BMR) over the course of their lifespan (LS) (reported maximum LS 9 years; average LS of several years) ([23]; F. Nottebohm, pers. comm with DH).

Laboratory studies using birds have obvious drawbacks as well. Investigations of the effects of experimental interventions on avian longevity in the lab have often relied on short-term observations of cohorts or subpopulations, rather than longitudinal studies, even though lifetime, longitudinal data remain the gold standard for any modelanimal species. Cohort-based studies often use short-term interventions and assays of putative biomarkers of aging and then extrapolate their results to longer-term outcomes. For example, repeated bouts of experimenter-induced stress in Eurasian blackbirds (*Turdus merula*) reportedly led to a decline in telomere length and an increase in markers of oxidative damage, both of which have been promoted as putative biomarkers of aging [24]. (Although this result led the authors to conclude that biological aging was accelerated in this model bird, survival rates were never assessed.)

Although it has become more common, the validity of using telomere attrition as a marker of differences in aging rates within or among bird species has not been firmly established. Telomere shortening and telomerase upregulation can vary widely even among species in the same vertebrate order, and also vary a great deal depending on developmental stage and the tissues being assayed ([25,26,27,28,29]; for a recent, nuanced perspective on the use of telomere shortening as a measure of stress or aging in bird studies, see especially [30]). Moreover, laboratory rodent studies have shown that inbreeding significantly lengthens telomeres, and that telomere dynamics can vary in a species or strain-specific manner [31].

## 3. Early-Life Interventions and in Ovo Developmental Studies

A large body of work in the field of aging has focused on how early-life interventions affect aging and longevity—for example, pre- and postnatal manipulation of caloric intake in various species (see, for example, [32]). For bird biologists, manipulation of the prenatal environment in ovo is an ongoing and intense area of study, due in large part to the relative ease of implementation and the separation of the developing embryo from more direct physiological maternal effects during development. For example, Japanese quail exposed to potentially stressful variation in ambient temperature in ovo have been reported to display altered telomere dynamics—again, consistent with the expectation that shorter telomere fragment lengths correlate with shortened lifespans. The inherent difficulty of using telomere dynamics as an indicator of aging were compounded in this study by the fact that telomere “aging” was measured in birds that were no more than two months old, well short of maturity or the median longevity for this species [33]. In another study, juvenile zebra finches subjected to simulated traffic noise also exhibited an increased rate of telomere loss, consistent with the expectation that the long-term survival of exposed birds would eventually be compromised [34]. Unfortunately, long-term survival is usually not assessed directly.

Nonetheless, there are recent examples of more longitudinal vs. cohort-based lab studies focused on putative aging or stress biomarkers. For example, a recent project involved treating zebra finches for over five years with a mitochondrial uncoupling agent, 2,4-dinitrophenol (DNP), via drinking water. This treatment reduced actual survival of the birds but, somewhat paradoxically, did notafect body mass, telomere dynamics, or markers of oxidative stress [35].

Most studies involving early-life intervention have focused on endocrine-driven effects on longevity—especially the effect of the steroid hormones corticosterone (CORT) and testosterone (T). CORT is the primary glucocorticoid secreted by the avian adrenal gland in response to stress and is critical in vertebrates for the maintenance of homeostasis during behavioral and physiological challenges, in addition to its role in metabolic regulation and maintenance of body condition [36,37,38]. Although currently a topic of great interest, the relationships between CORT level, telomere length, and markers of oxidative stress and longevity in birds remain unclear. For example, no apparent relationship between CORT and harsh conditions was detected in recent studies of starlings [39] or chickens [40], while Haussman et al. [41] reported a decline in telomere length in domestic chickens given CORT in ovo (but the effect of CORT level on survival was not assessed).

In a similar vein, Monaghan et al. [42] reported that early-life stress elevated hypothalamic-pituitary-adrenal axis activity in zebra finches, and that an increase in CORT was associated with a shortened lifespan, but there was no overt effect on reproduction. A similar effect on mortality patterns has been reported for the house sparrow, independent of biochemical markers of health [43]. In that study, survival for the first four years of life was negatively associated with variance in body mass, such that smaller house sparrows showed reduced survival rates. This differs from the pattern typically seen in mammals [44], wherein larger species tend to have lower mortality rates and longer lifespans. One recent study suggested that individual variation in basal CORT level—but not stress-associated levels—was an important, sex-specific determinant of survival in zebra finches [45]. On the other hand, another group reported that young adult zebra finches demonstrated a beneficial effect of repeated stress on survival [46]. Because of the significant role played by CORT in carbohydrate metabolism, it wouldalso be reasonable, at least tentatively, to explore longevity-associated differences in resting blood glucose among zebra finches that may be affected by CORT level [47].

Additional evidence for an association between prenatal hormone treatment and avian longevity has been provided by studies manipulating sex steroids early in development. A decade-long follow-up study of in ovo T-treated homing pigeons (*Columbia livia domestica*) showed that females, but not males, showed a significant reduction in survival [48]. (The significance of this finding is tempered by an extremely small sample size: *n* = 4–6 birds per treatment). Schwabl et al. [15] also reported a sex-specific effect of prenatal T treatment on adult mortality patterns in captive house sparrows. In contrast, young adult (one-year-old) male red-legged partridges (*Alectoris rufa*) implanted with flutamide (an androgen receptor antagonist) or flutamide plus 1,4,6-androstatriene-3,17-dione (an aromatase inhibitor), showed no difference in median survival relative to that of untreated controls. Mortality was accelerated in a T-treated group in this study, but unfortunately an avian *Escherichia coli* enteritis outbreak confounded interpretation of these data [49].

Experimental studies of avian physiology with implications for health and longevity, such as those summarized above, are still relatively rare. The papers cited here represent a very small body of work relative to the hundreds of reports each year featuring more traditional short-lived models of organismal aging and disease, such as laboratory rodents. This likely reflects unfamiliarity with the potential of avian models on the part of the basic aging research community, as well as apprehension about the challenges associated with the husbandry of an unconventional animal model. There is also a lack of communication or intellectual integration among the fields of comparative geroscience, avian physiology, and evolutionary biology. Notwithstanding these issues, it has been argued for more than three decades that birds are an underappreciated model organism for aging research (e.g., [3,4,5,6,7,8,50,51,52,53]). Although studies claiming to show age-related changes in a variety of biochemical, physiological, and anatomical measures in birds have been conducted for some time, the majority of these have not been aimed directly at basic mechanisms that are established correlates of vitality, reproductive success, aging rates, or lifespan variation. More recently, however, progress has been made in developing additional laboratory approaches to studies of aging using birds, particularly in established cell-based protocols for studies of mammalian aging.

## 4. Primary Cell Line Culture

Significant momentum has been established recently in studies using primary cell lines cultured from vertebrates with disparate lifespans; these have been aimed at dissecting basic mechanisms underlying the dramatic variation in aging rates among species, strains, and populations. Initially, these studies tended to focus on the relationship between cytotoxic stress resistance and species maximum lifespan (MLS) [50,51], a rough measure of resistance to aging-related mortality. However, with the advent of real-time metabolic assays, we can now more directly examine the relationship between cellular metabolism and variation in individual longevity within a species (for examples, see [53,54,55,56,57]). These studies have relied heavily on fibroblasts, a cell type readily obtainable from any organ that has a large proportion of connective tissue. But skin is the most common source because it is abundant, easily biopsied, and lends itself to studies requiring minimal invasiveness. If an animal is euthanized prior to sample collection, many cells can be generated in culture within a few days [58], although skin samples can be collected from living individuals using punch biopsies, ear punches, or toe or tail clips (in mammals), and a single individual can be sampled repeatedly in longitudinal studies. Fibroblasts are one of the easiest cell types grown in culture by far, and they are amenable to a wide range of culture conditions (in fact, that they are a common “contaminant” in primary cultures of other cell types). Once established, individual cell lines proliferate rapidly in culture, and can generate tens of millions of cells for testing within several weeks.

Another advantage to this approach is that the basic methodologies are transferable from one species to the next with some taxon-specific modification of techniques (such as the inclusion of sodium pyruvate and chicken serum in avian cell culture media). Fortunately, mammalian fibroblasts are readily grown in standardmedia as well, allowing for cells from multiple species to be grown and tested in parallel, and facilitating comparisons between taxa.

In a comparative study of mammalian species, a strong relationship between cellular stress resistance and organismal longevity was established using primary dermal fibroblasts derived from long-lived mutant mouse strains (i.e., Snell and Ames dwarf) and growth hormone receptor knockout (GHRKO) dwarfs [59,60,61]. Briefly, cells derived from long-lived rodent strains were shown to be resistant to the lethal effects of an array of cytotoxic agents [62,63,64]. Notably, each of these early studies only compared cells grown from long-lived genetic mutants relative to shorter-lived wild-type mice, rather than broader taxonomic groupings of mammals or birds. But more recently these findings have been shown to apply to other mammalian species as well: cells from longer-lived species have been shown to be more resistant to cytotoxic stress, even after controlling for the effects of body size and phylogeny [65,66,67].

In a similar vein, primary cell lines derived from a wide range of avian species have been used to explore the mechanisms underlying their exceptional longevity. An initial study ofstress resistance inrenal cells from budgerigars (*Melopsittacus undulatus*), starlings (*Sternus vulgaris*), and canaries (*Serinus canaria domestica*) demonstrated that avian cells were more resistant to oxidative stress and DNA damage than a typical inbred, short-lived laboratory mouse cell line [68,69]. Cells grown from short-lived Japanese quail (*Coturnix domestica*) were also shown to be less resistant to stress than cells grown from the much longer-lived budgerigar (*Melopsittacus undulatus*) [70]. However, a more recent study using embryonic fibroblasts from a short-lived (domestic chicken) vs. a long-lived bird species (pigeon) failed to show any appreciable difference in stress resistance. A later, more extensive comparative survey utilizing fibroblast cell lines from eight avian orders and more than 30 species found a significant correlation overall between cellular stress resistance and species maximum reported lifespan (MLS) [71].

An established principle in comparative zoology and evolutionary biology is that within closely related taxa, individual species fall along a continuum in terms of “life history traits” (demographic vital rates), including maturation rates, reproductive output, lifespansaging rates. This continuum is postulated to reflect demands imposed by ecological space, mortality rates from predation and other extrinsic causes [4,7,72,73,74,75,76]. According to this “slow-fast” life history paradigm, the “slow” end of the axis is characterized by relatively slow development and maturation, low reproductive output, low mortality rates, and short lifespans. The “fast” end, on the other hand, shows an association between fast maturation, high reproductive output, high extrinsic mortality rates, and rapid aging. Potentially useful comparisons of model organisms that display slow and fast aging can be made by exploring the extremes of this continuum [5,6,7,74]. Trade-offs between “slow” and “fast” life histories are predicted to reflect responses to the environment and the forces of natural selection acting on the genome that ultimately shape the life history and other phenotypic traits, including cellular biochemistry, metabolism, and other aspects of a species’ “physiological architecture” [74]. In support of this scenario, there are well documented differences between life history patterns in bird species from temperate vs. tropical zones. Tropical species generally tend to grow more slowly, exhibit lower metabolic and reproductive rates, and have lower mortality rates than temperate-zone species. These differences have been suggested to reflect variation in extrinsic mortality force (e.g., from predation or infectious disease) and environmental stability [7,72,73,74,75,76].

When the stress resistance properties of cultured fibroblast cell lines derived from pairs of north temperate vs. neotropical species were compared in the laboratory, the results wereconsistent with this notion. Cell lines from “fast” (i.e., shorter-lived) temperate birds were significantly less resistant to an array of stressors than were cells from phylogenetically matched “slow” tropical species [75,77]. Moreover, cellular metabolic rates were shown to be higher in cells from “fast” temperate species using multiple measures (e.g., O_2_ consumption rate, proton leak, and the rate of anaerobic glycolysis) [55].

Despite an association between cellular stress resistance and longevity being generally well supported both within and among mammalian and avian taxa, the biochemical and physiological processes underlying this relationship remain elusive. Notably, recent laboratory work using an extremely long-lived, slow-aging animal, the naked mole-rat (*Heterocephalus glaber*), has revealed changes in proteasome activity at advanced ages [78,79]. A finding of improved molecular repair capacity extends so far to cell lines of other long-lived mammals, as well as long-lived birds, relative to shorter-lived species [80,81]. Other mechanisms implicated in studies comparing these taxa include altered kinetics of MAPK-(mitogen-activated protein kinase-dependent) signaling, a key mediator of stress response pathways [82] and DNA damage repair [80]. The finding of improved repair capacity in long-lived species is consistent with exceptionally robust damage repair and differences in cytotoxin resistance among cell lines from long- vs. short-lived strains of mice [83]. Other clues to the protective mechanisms distinguishing longer-lived animals have come from examining membrane lipid composition (for a review, see [7]; see also [84,85,86]), as well as metabolic properties of avian cells in comparison to cell lines derived from mammals [55,80,86]. Finally, altered constitutive expression of NRF2 (nuclear factor erythroid 2-related factor), theconsequence of a mutation in Kelch-like ECH-associated protein 1 (KEAP-1) conserved within class Aves, has been proposed as a major factor in the evolution of long lifespan [87]. This, too, is consistent with findings for long-lived non-avian species [88]. Taken together, the existing body of evidence supports the idea that there are strongly conserved anti-aging mechanisms within and among homeothermic vertebrate classes, including Aves.

## 5. The Erythrocyte Model

Mature mammalian erythrocytes (red blood cells) are unique among vertebrates in that, with few exceptions, they lack nuclei and other organelles (i.e., endoplasmic reticulum and mitochondria). These components are unnecessary for the transport of oxygen and carbon dioxide, and theyare actively extruded from the mammalian red blood cell during development via an autophagy-dependent process [89].

They are not physiologically inert, however, and mammalian erythrocytes have utility for evaluating many aspects of whole-animal physiology and can serve as an important diagnostic tool for certain pathophysiological states, e.g., elevated levels of glycated hemoglobin (A1c), a clinical indicator of poor glycemic control. (In a possible exception to this rule, a recent study proposed that variation in rates of erythropoiesis among mammalian species may be correlated with lifespan potential [90].) Avian erythrocytes, on the other hand (similar to fishes, most amphibians, and non-avian reptiles), retain a nucleus and functional mitochondria [91,92]. This has stimulated a plethora of studies focused on erythrocyte characteristics as an ostensible marker of aging in birds, including both wild or free-living populations and domesticated species like zebra finches [7,92]. Blood samples are readily drawn from a wing vein, even from smaller bird species like finches or sparrows, leading to the widespread use of avian erythrocytes or whole-blood samples to investigate putative mechanisms of agin. This is due to the relative ease of sample collection, as well as minimal invasiveness and adaptability by bird biologists in lab or field. This approach is particularly attractive for longitudinal studies requiring repeated sampling of populations monitored using mark-recapture methods.

One popular rationale for using avian erythrocytes is that they are a convenient tool for assessing oxidative risk and potential damage to the whole organism, at least in a snapshot of a certain developmental stage. But whether oxidative stress is a causal or correlative factor in vertebrate aging in general is still much debated in geroscience. There is substantial evidence for age-related changes in oxidative burden or antioxidant defense systems in several model systems, however. If valid, this has implications for both endogenous and exogenous sources of oxidative stress [93,94]. In published bird studies, the endogenous production of mitochondrially-derived reactive oxygen species (ROS) within erythrocytes is widely hypothesized to contribute to the oxidative burden for the organism as a whole, as is the production of ROS from other tissues [91,92]. In addition, erythrocytes (or more typically the cellular fraction of whole blood samples) can be isolated and assayed for benchmarked biomarkers of oxidative stress such as DNA damage (e.g., 8-oxo-2’-deoxyguanosine, 8-oxo-dG), lipoperoxidation products from cell membranes, or protein carbonylation. Other measures, such as the time to hemolysis in the presence of oxidizing agents, are also used as a de facto measures of antioxidant capacity.

Often such measures are coupled with an assay for overall or total antioxidant capacity (AOC/TAC) of plasma or whole blood using a test that employs hypochlorous acid as an oxidizing agent. Despite their extensive use in the bird literature [95,96,97,98,99,100,101,102,103,104,105,106,107,108], the data generated using isolated erythrocytes and/or blood AOC have been equivocal in terms of making strong inferences at the whole-organism level, butconsistent with some studies using controlled experiments to alter oxidative measures (see, for example [108]). Much of this lack of consistency is likely due to poor standardization of assays among laboratories or an inability to account for numerous confounding factors, such as interspecific variation in blood protein composition (but see [93]). On the other hand, the antioxidant properties of erythrocytes may be a useful, if limited, indicator of early life events and interventions [49,109,110].

Other oxidation-based assays, such as quantification of heme degradation products [111] or DNA-breakage detection fluorescence using in situ hybridization [112], have not been explored in the context of aging and longevity, but do show promise. On the whole, few studies acknowledge that little direct correlation has been established between markers of resistance to oxidative damage, other stressors, vitality, long-term reproductive output, or aging-related pathophysiology; and there is often an unstated presumption that variability in erythrocyte physiology is indicative of the functionality of other cell types. This is currently not confirmed by direct inference [7].

In addition to investigations focused on measuring oxidative stress or damage, the rate at which non-oxidative DNA damage accumulates in leukocytes of some mammals and birds has been shown to significantly correlate with species lifespan (MLS) [113]. One group reportedthat the accumulation of a marker of DNA double-strand breaks, the phosphorylated histone H2AX (γH2AX), was generally lower in long-lived species, especially birds. In addition to prevention of damage, the maintenance of genomic stability downstream of DNA damage repair capacity has been implicated in lifespan regulation (for a review, see [114]). Whether this stability also holds true for erythrocytes has not been assessed. Avian erythrocytes have largely been ignored for studies of DNA damage, except in the context of xenobiotic toxicity [115] or as a marker of the stress response via the hypothalamic-pituitary-adrenal axis [116,117], despite its potential as a marker of variation in aging processes [118].

Despite these uncertainties, because they are nucleated avian erythrocytes are a ready source of genetic material; usable quantities can be collected in little more than a few drops of blood [119]. Henc this blood component does have potential for comparative genomic studies of short- versus long-lived species, and may prove especially useful for species that are of concern for conservation biologists. For example, the red-crowned crane (*Grus japonensis*) is a traditional symbol of longevity in East Asia due to its exceptional lifespan (up to 65 years in captivity [120]). It is also an endangered species. Notably, Lee et al. [120] were able to perform whole-genome sequencing using a blood sample collected from a single individual crane (with analysis of the genome in reference to other avian species), and tosuccessfully identify pro-longevity candidate genes, especially those involved in metabolic and immune function. Likewise, parrots (order *Psittaciformes*) are particularly long-lived relative to other similar-sized birds [121], with some species living more than six decades [122]. Using a similar approach to that in the crane study, whole blood samples have been used to generate genomes from multiple parrot species and have revealed several genetic features unique to this long-lived group, as well as other features that are common among long-lived birds in general [122,123]. In addition to the finding that metabolic and immune related-genes can produce a strong genetic signal in species with exceptional longevity, these studies have detected genetic signatures of improved cellular defense mechanisms and differences in the regulation of pathways involved in cell growth and proliferation.

Avian erythrocytes and their functional mitochondria also lend themselves to mitogenomic studies whose focus is the genome of isolated mitochondria, rather than the whole genome. This approach has revealed that the avian mitochondrial genome has some unusual properties compared to that in other vertebrates in terms of structure and organization [124]. For example, the avian mitogenome possesses bird-specific features, including gene duplications and an expansion of the mitochondrial control region shown to be associated with increased longevity [125,126]. Moreover, the degree of mitochondrial genetic heterozygosity may account, at least partially, for longevity variation among individuals within a population [127]. A strong association has been detected between mitochondrial parameters and interspecific variation in the MLS of 21 bird species [128].

Erythrocytes have also been used for environmental epigenomics studies designed to reveal changes in DNA methylation patterns associated with prenatal rearing conditions and temporal variation in temperature [129,130]. Epigenetic regulation of numerous processes is thought to be a key factor in the aging linking longevity to diet, metabolism, and disease processes [131].

Within the past decade, hydrogen sulfide (H_2_S; and more specifically, the transsulfuration pathway), has generated considerable interest as a modulator of the established beneficial effects of caloric restriction on aging and longevity in rodents [132], as well as an intervention for extending the lifespan in other model organisms downstream of H_2_S-mediated signaling [133]. We (JH’s lab) have proposed a novel use of erythrocytes for the study of avian aging: i.e., the quantification of transsulfuration pathway activity among short- versus long-lived species, or in response to putative anti-aging interventions using blood samples (in accordance with [134]). This is made possible using avian blood by the presence of functional, active mitochondria, plus the finding that, at least in chicken erythrocytes, H_2_S is actively produced and plays a prominent role in cellular bioenergetics [134].

## 6. The “-omics” Era and Studies of Avian Aging

With the advent of automated, high-throughput technologies and expandeded computing power, the last several decades have seen a monumental growth in studies whose focus is on various levels of biochemical organization within cells, tissues, and organs of species from virtually every taxonomic order. These include genomic (nuclear, mitochondrial, chloroplast), transcriptomic, proteomic, lipidomic, glycomic, and metabolomic datasets, as well as microbiome data generated under varying experimental paradigms. Moreover, specialized datasets such as the kinome (the array of protein kinases encoded in an individual genome) and the methylome (the set of all nucleic acid methylation modifications within a genome), are also being produced. Not surprisingly, each level of -omic organization has been studied in the context of organismal aging to reveal suites of candidate genes, etc., putatively involved in the aging process. However, these studies have mostly been limited to the classic model organisms used in basic gerontological studies: yeast (*Saccharomyces*), nematodes (*C. elegans*), fruit flies (*Drosophila*), and mice (*Mus*). We focus in the following sections on -omics data gleaned from the use of avian models that are potentially relevant to geroscience; we also highlight how data from birds compare to data collected using more “traditional” animal models.

### 6.1. Avian Genomics

As noted above, when using erythrocytes as source material, several groups have demonstrated that the genetic signature of long-lived birds shows an enrichment of genes involved in stress defense mechanisms, cell growth and proliferation, and immune function. This agrees generally with data now available from other species with exceptional longevity [135,136], and from studies of long- versus short-lived individuals within species, including humans [137,138,139,140,141]. Genomic profiling has also proven to be a robust predictor of human aging [142]. However, avian genomics is still in its early stages, with most of the data having been generated within the last five years, since whole-genome sequences have become widely available only recently [143,144]. Publishedstudies have either focused largely on phylogenetic relationships among avian taxa or specific hypothesis-driven investigations into the evolution of specific loci (e.g., the major histocompatibility complex (MHC) [145]). It isnoteworthy that the copy number of MHC Class I genes in birds has been shown to be positively associated with longevity [146].

The first detailed analysis comparing bird genomes using a hierarchical approach to examine genetic convergence for life history traits has just been published [147]. There are few studies of avian genomics that are aging-centric; hence, birds and non-avian reptiles still represent an untapped resource for gerontological studies [148]. As interest in using birds for aging research grows, so too will our knowledge of the genetics of avian aging. It is also noteworthy that the avian genome is unique among vertebrates in a number of ways—most notably, adrastic reduction in size [149]. In addition, interest in the mitochondrial/mitonuclear genome and its role in aging has increased appreciably in recent years [150], and shows that, once again, birds exhibit some unique features, especially in terms of mitochondrial genome size and organization (see again [125,126]). However, as with whole-genome data, relatively little is known regarding the functional genomics of avian mitochondria [151].

### 6.2. Avian Transcriptomics

Even though genomic data are a useful tool for examining the genetic architecture of a given species or individual, they often do not provide information about which genes are expressed and where (i.e., which tissue/organ); the magnitude of expression; or the development stage(s) at which they are expressed. Ultimately, the generation of organ-specific transcriptomes will be possible at various points in a model organism’s life history, as well as under specific experimental conditions, including anti-aging interventions As the field has grown, the human aging transcriptome has been shown to be a robust tool for estimating the biological age of an individual [152], as well as measuring age-related changes in tissue-specific transcriptomes of model organisms such as monkeys [153], mice [154], *D. melanogaster* [155,156], *C. elegans* [157,158], and yeast [159]. Not surprisingly, relatively little is known about the aging transcriptome in birds, with the exception of studies of brain [160] and thymus [161] transcriptomes in populations of domestic chicken. Dramatic differences in the basic genomics of wild vs. domesticated species may be problematic when trying to extrapolate findings from these studies to birds in general [162]. Results of at least one study that examined the transcriptomes of male versus female magpies (*Pica pica*) in captivity suggested a sex-specific differential in expression of genes associated with longevity, especially those associated with stress resistance and the sirtuin signaling pathway [163].

### 6.3. Avian Proteomics, Lipidomic and Metabolomics

Information about higher -omic levels in avian models of aging is scarce, and it is not surprising that what little is known comes from studies involving muscle tissue of agriculturally relevant species, such as chicken and duck [164,165]. One notable exception is a comparative study in which fibroblasts were cultured from a group of avian species and grown in parallel to those from other long-lived animals, such as bats [81]. Ma and colleagues found that the proteomic and metabolomic signatures of individual cell lines appeared to be conserved among longevous species, including birds [80]. This is consistent with similar data from whole-animal models, including Ames dwarf mice [166], bats [167], and naked mole-rats [168]. The proteomic and metabolomic signatures were also consistent with those from studies employing life-extending interventions, such as caloric restriction [169]. Serum proteomic profiles have been shown to be a robust predictor of longevity in humans [170].

The lipid composition of avian plasma membranes has also received some attention, in accordance with the “lipid membrane pacemaker theory of aging” [171,172]. These data generally support the notion that a reduction in the degree of polyunsaturation of membrane fatty acids typical of avian cells may contribute to birds’exceptional longevity ([85] and references therein). While there is growing interest in the gut microbiome and its influence on organismal metabolomics and physiology [173]; it has received no attention for avian species apart from domestic birds. Clearly, much remains to be investigated with regard to each of these areas, and the data integrated further with what is already known about the regulation of avian aging and longevity at physiological, cellular, and molecular levels.

## 7. Neural Aging and Cognitive Function in Birds

Ottinger [16] has reviewed studies of avian neurodegeneration and aging-related changes in cognition. Here, we summarize some key points from earlier studies, while also highlighting developments in this area in the past several years. Although in most animal models there is an absence of the particular brain pathology seen in Alzheimer’s disease (reviewed in [1]), aging-related declines in cognitive function are well described in various animal models, including *D. melanogaster* [174], *C. elegans* [175], and in many respects resembles the changes seen in humans. Birds appear to be no exception: most avian species studied to date show some degree of impaired neural function and cognition as they age. In any case, there are still relatively few published studies of aging-related changes in neural function and cognition in birds overall, including studies of age-related neurodegeneration.

Since the time of the ancient Egyptians, the homing ability of the pigeon (*Columba livia domesticus)* has been exploited—first as a messenger system and later, for sport enjoyed by hobbyists all over the world. So it is not surprising that domestic pigeons have been the subject of numerous studies of the neurophysiological bases of orientation and related processes, with many focusing on vision. Pigeons (and, likely, other species of birds) exhibit declines in visual acuity due to the loss of retinal ganglia as they age [176,177,178,179,180]. These deficits are likely exacerbated by deficiencies in choroid flow dynamics with advancing age [181,182,183]. Studies of Japanese quail have also shown a significant reduction in contrast sensitivity with age, although other aspects of their vision havenot shown significant changes [184]. Both pigeons and Japanese quail exhibit age-related declines in photoreceptor number and a redistribution of retinal microglia [185]. In the common kestrel (*Falco tinnunculus*), marked age-related changes in the degree of pigmentation, levels of key neurotransmitters, and the structure of the optic nerve with advancing age have all been reported [186].

In contrast, auditory function is relatively unaffected with age in both starlings (*Sturnus vulgaris*) [187] and barn owls (*T**yto alba*) [188]. (Interestingly, this phenomenon has not been studied in pigeons due to technical difficulties associated with this species [189]). A remarkable regenerative capacity of avian hair cells is believed to underlie the maintenance of auditory function in aging birds [190].

In addition, spatial memory and learning both decline with age in the pigeon, despite paradoxical increases in brain mass and hippocampal volume [191,192,193,194]. Since the hippocampus plays a dominant role in avian learning and memory formation ([195] and references therein), it is not surprising that dysfunction of the hippocampus and related brain structures, such as the septum, are believed to be responsible for the observed declines in spatial memory and learning regardless of the maintenance of brain mass. In particular, hippocampal dysfunction is the result of impaired neural activation [196] or a reduction in the degree of neurogenesis in specific hippocampal regions ([197]; also see below). This is true as well for impaired neurogenesis in the olfactory bulb, which plays a prominent role in pigeon homing [197]. Similar declines in learning and memory have also been reported in Japanese quail [16].Interestingly, however, normal cognitive function in both parrots and zebra finches persists well into adulthood, perhaps as a result of differential gene expression related to longer lifespans [198].

Song is an integral part of life for many birds, male and female, and an essential form of communication during the breeding season. A vast literature exists dedicated to song acquisition and maintenance in a wide variety of avian orders and species, collected using various experimental paradigms. Along with the canary, the domestic zebra finch has been a dominant model for the neurobiology of song (most of this literature is beyond the scope of this review; refer to [199] for more information). On a proximate level, song acquisition is under the control of steroid hormones in the brain, especially estrogens [200,201,202]. These are regulated, in turn, by the target of rapamycin (mTOR) metabolic signaling pathway [203]. mTOR is a prominent player in the regulation of aging across multiple model taxa, making this finding especially intriguing [204].

Research conducted to date suggests that individual birds’ song repertoires are retained as the organism ages, most likely due to the maintenance of an independent memory system that governs song separately from other processes, such as spatial memory [205]. Integral to this process is a highly neurogenic region of the brain known as the high vocal center (HVC) [206,207], which is characterized by constant neuronal replacement. This can occur in birds of all ages, but the pace changes with age or, more specifically, the rate of incorporation of new neurons declines appreciably as birds age, leading to the accumulation of “old” neurons within the HVC [206,207]. A recent study has demonstrated that expression of activity-dependent neuroprotective protein (ADNP) also declines with age in birds, although not specifically in the HVC [208]. ADNP is essential for normal growth of new dendritic spines and brain formation in mice [209]; the loss of expression with age may contribute to neuronal loss. In addition, some aspects of a bird’s song, such duration or intensity, appear to undergo a significant change with aging. Consequently, age-related changes in features of species-specific song have been proposed as a useful marker of functional senescence in wild bird populations, since the data required to monitor this would be relatively easy to collect [210]. Interestingly, the changes in song composition typical of an aging bird parallel those seen in the speech pattern of aging humans and are associated with altered vocal motor control [211,212,213,214,215].

## 8. Future Directions and Concluding Remarks

The promise avian models hold for the search for longevity-assurance pathways has now been appreciated for several decades. Despite their extremely small size and extraordinarily high mass-specific metabolic rates, even hummingbirds can live up to a decade or more, and were proposed as a model for studying longevity in the wild as early as the 1980s [215]. However, there has beensurprisingly little aging research involving birds since then.

The past decade, however, has seen a dramatic uptick in the number of studies of avian aging, both in the laboratory and in the wild. Unfortunately, the focus of these studies has often been limited by the underlying supposition that variation in aging rates among species or populations are driven primarily by differences either in oxidative stress resistance and damage or by changes in telomere dynamics. While the accumulation of oxidative damage [215] or dysfunctional telomere dynamics [215] have been shown to be involved in pathophysiological processes in some model organisms, in bird studies (as in human studies), there is a lack of consistency in the methods used to examine these processes. There is great variability in the validity of biomarkers used, as well as the in the strength of inferences drawn from many of these studies [7], as we have emphasized in earlier reviews. There is no stand-alone biomarker or cellular “clock” that reliably reflects variation in the rate of aging among diverse organisms.

The adoption of high-throughput -omic tools by avian biologists interested in geroscience holds a great deal of promise for the future. As these technologies have become increasingly economical, they are no longer the exclusive domain of medical schools or Tier 1 research institutions, but are accessible and affordable to investigators at modest-sized colleges and universities. There are also excellent opportunities for field biologists as collaborators at smaller institutions. Thesecould open opportunities for hypothesis-driven, interdisciplinary, and well executed -omics-based studies that were not feasible as recently as a decade ago. Studies integrating established approaches from field ornithology are attractive to many undergraduate and masters-level students, moreover, and their involvement can help push the field forward as well. We are hopeful that these factors will facilitate progress in the area of avian aging in the coming years, as well as stimulate comparative investigations into mechanisms of aging in other poorly studied long-lived species of species of special gerontological interest (e.g., turtles and other non-avian reptiles) that have been heretofore neglected.

## 9. Summary

Despite the fact that class Aves (the birds) is an exceptionally long-lived group for their generally small body sizes and high metabolic rates, the use of birds as model organisms in aging research is still in its infancy relative to other vertebrates (especially rodents). Here, we have provided an overview of recent progress using birds in the laboratory for studies of aging and aging-related processes. We have highlighted some relevant and relatively recent studies of aging-related phenomena, which have used wild as well as domestic birds. We have traced the development of cell-based models as an alternative to whole-animal studies, as well asprogress in identifying genetic and cellular correlates of avian longevity. We have also discussed recent studies based on aspects of gene expression patterns, as well as aspects of neural aging and cognitive function, that may provide new perspectives on the molecular bases of aging in birds and other long-lived vertebrates.

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
