# Peer review of "New Perspectives on Avian Models for Studies of Basic Aging Processes"

_biomedicines, 2021, doi:10.3390/biomedicines9060649_

Round 1

Reviewer 1 Report

My comments are included in the attached file.

Author Response

General Comments

I think this is an interesting and timely topic in the biology of aging, and the authors are to be commended for their effort in presenting the available literature. However, unfortunately this seems to be a somewhat untidily written and occasionally self-contradictory review that would benefit from a more rigorous progression of points and arguments. There also seem to be quite a few typos and grammatical oddities, and the manuscript would benefit from being thoroughly proofread before resubmission.

I think the authors’ framing of mechanisms involved in the aging process is a bit too simplistic – while they show the seminal figure from Lopez-Ortín, they are far from addressing every mechanism discussed in that paper in theirs. There needs to be a justification as to why the authors choose the particular mechanisms that are part of their present review.

Abstract

Lines 21-22, this sentence seems to be out of order

Thank you for your comment.  The wording has been changed in the section and it has been rewritten.

I do not understand the authors’ rationale to single out oxidative stress and telomeres as mechanisms of aging here, please clarify.

Introduction

Line 47: What do the authors mean by “taxonomy” here? Please clarify.

Section has been substantially reworked and should be clearer now. Obscure us of “taxonomy” was deleted.

Line 62 and following should also reference the domestic dog, the rhesus macaque, and the African killifish as newly emerging nontraditional models of aging.

The rhesus macaque has been a model animal in aging studies for over 30 years now. We didn’t intend to list all animals used in gaining studies, but to highlight the use of birds and contrast them with some other traditional models. The killifish was noted as a novel short-lived model.

Lines 80/81: Do not use contractions in academic papers.

Of course not. It’s gone now.

Avian Aging Studies in Field vs. Laboratory

Line 85 ff, could you clarify to what degree these models took an exponential increase in the risk of death over time into account? This is a commonly used definition of aging in actuarial studies.

We appreciate the reviewer’s comment, but this information is not typically available in published studies.  Each of the studies cited was a review and there are no hazard rate data presented. There are a number of wild bird populations for which actuarial aging patterns have been described very well. We refer back to our reviews in the 1990’s – very long-lived wild birds are probably not great emerging animal models – we have chosen to focus on some more promising species and approaches.

Line 107, not sure I would agree with this statement given the significant body of literature involving rhesus macaques.

First, the MS is about birds as aging models and related studies. It wasn’t intended to review and all vertebrate models.  We have tightened up the Intro to identify our objectives better. The reviewer is correct that there are several long-term studies involving rhesus macaques, one at the NIA and one at the University of Wisconsin.  These are the source of most of the literature involving this species and are unique in their design and duration.  In comparison to other model organisms, studies involving long-lived species are generally the exception not the norm.   (and as noted above, monkeys have been used in aging studies for over 30 years now, so they are not new.

Line 109, this appears to be changing, notably with regards to the increasing number of aging studies using the domestic dog.

With the exception of studies of cognitive aging, most studies involving dogs that we are aware of take a comparative approach that relies on cross-sectional studies of short- versus long-lived breeds.  The desire for shorter-lived comparative models stems from the desire to conduct interventional studies in species other than mice.  To our knowledge, there have only been a handful of dog longevity studies utilizing nutritional interventions that were funded by private entities.

One of us has just enrolled our young dog in the big Dog Aging Project designed by Promislow and Kaeberlein at U of Washington. We should know a lot more about aging in dogs in a few years.

115 and following, while I can see the argument for using sparrows, it does seem that their lifespan as stated by the authors does not make them particularly long-lived for a vertebrate in their body weight range, compared to e.g. a mouse. In addition, the lifespan the authors give for domestic quail would appear to put them in a shorter-lived category considering their body weight. This needs to be acknowledged given that the authors are also arguing that the exceptional longevity of certain bird species is what makes them potentially interesting models in geroscience.

We appreciate the reviewer’s comment and have edited the text to address their concern. There is little data available on effects of in ovo T treatment on wild songbirds. We could obviously not assess aging per se in this study, but rather we described sex-specific  mortality patterns over a period of several years. We had a small sample size, but since this is the only study really looking at sex-specific changes in a wild songbird’s mortality patterns in captivity after prenatal T treatment, we’ve cited it. Wording has been adjusted so as not to make false claims.

Line 141 ff. it would be interesting to briefly discuss the potential pitfalls of using telomeres in mice in particular as an example of this phenomenon.

A brief discussion has been added to the text.

Early-life interventions and in ovo developmental studies

Line 151, parenthesis needs to be closed

This has been done

In the first paragraph, it is interesting to read about studies of in ovo interventions affecting telomere length; however, the authors should point out more clearly that a correlation between telomere length and actual lifespan and/or aging in birds is not sufficiently well-established to conclude that these indicate aging.

We believe that the paragraph, as written, clearly indicates that these data are inconclusive for several reasons.  That telomere length is an unreliable indicator of aging has already been covered in the preceding section.

Line 173 and following, do we know whether these decreased lifespans are associated with increased mortality risk across all ages, or are they specific to an earlier increase in exponential mortality risk over time? The latter would present a much better argument for an actual effect on aging than the former for the reasons discussed in my comment for Line 85 ff.

Again, we appreciate the reviewer’s comment and agree with the argument being put forth.  These data are not readily available outside of a visual inspection of the survival curves (when provided).  Consequently, we don’t feel it’s appropriate to address this issue in the manuscript in the absence of an analysis of the raw data.

Line 198 and following: To what degree does the karyotype of male/female birds interact with these findings? Do male birds generally live longer than female birds or vice versa?

The literature is limited, but there is either no difference among the sexes or a tendency for males to be longer-lived (reviewed in Austad and Fischer 2016).  However, we chose not to address this in the text because it would not change the interpretation of the reported outcomes.  

Primary Cell Line Culture

This seems to focus mostly on cell culture in a broad sense and/or in mammalian models. The focus of this paper is avian, so I would recommend that the authors shorten the introduction to this paragraph. Also specify what kind of cells exactly were grown in the bird studies the authors cite.

We have chosen to leave the discussion of the mammalian literature as a reference for readers interested in this approach, as well as to reiterate the rationale for this methodology.  The cell lines used were noted where missing.

Line 298, please provide references for this statement.

References have been provided

Line 301, again, please specify what kind of cells.

This has been done

Line 307 and following, it might also be worth citing Steve Austad’s work on Arctica islandica cells in this context.

We have chosen to exclude Steve’s work on clams because it is an invertebrate model. 

The Erythrocyte Cell Model

Change the title of this subsection to “the erythrocyte model” to avoid redundancy.

This is a good point.  We have made the suggested change.

Line 370, reference list needs to be edited

Donna, I’ll leave this one to you.

Lines 374-378 this statement does not seem to logically belong in the place it is found at.

We have changed the wording to make it more clear why these statements have been included here.

Line 403, please provide the lifespan of this species. Same for line 409, please provide some concrete examples.

These have been added to the text.

Line 418 and following, there is a substantial amount of research on animal models of mitochondrial dysfunction. Have any such dysfunctions been described in birds?

In addition to the studies cited, the literature regarding mitochondrial function in birds has been limited to comparing mitochondrial energetics in birds versus mammals and not of mitochondrial dysfunction per se.

The “omics” Era and Studies of Avian Aging

Line 450, should also include references to the methylome and the microbiome, which have been described as being involved in the aging process.

These have been acknowledged

Line 482, please provide some concrete numbers to put these statements in perspective.

It’s not clear which statements the reviewer is referring to.  Nevertheless, we feel that a generalization stating that avian genomic studies are limited in number is sufficient.

Neural Aging and Cognitive Function in Birds

Line 539, either expand how poor cognitive function is not detrimental or remove this statement, as it does not appear to be particularly relevant to aging.

We agree that this statement was unnecessary and have removed it.

Have there been any avian studies of proteins known to be related to age-related neurodegenerative disease in mammalian models, e.g. amyloid-beta 42, phospho-tau, alpha-synuclein etc.? Please clarify if so, or that such studies are lacking if not.

There are no comprehensive studies of age-related neurodegeneration in birds that we are aware of.  This has been noted in the manuscript.

Future Directions and Concluding Remarks

The hummingbird example belongs into the introduction and/or the oxidative damage sections, it seems out of place in this section.

We feel inclusion this example here reiterates that despite a long-standing interest in avian aging, there has been relatively little done.

Summary

Please use bold font for this section title.

This has been done

This section could do with a somewhat more fleshed-out and succinct conclusion, emphasizing the authors’ point that birds are a promising and underutilized model in geroscience.

I’m indifferent on this point, Donna.  I think it’s fine as is, but I am happy with any changes you think are warranted (or not).

Reviewer 2 Report

In Harper and Holmes, the authors review what is known about aging in birds, and the potential use of Aves as models for aging research. Overall, I found the paper interesting, and I think it is a needed manuscript in the field. I do not have any major issues, just a couple questions/comments, listed below in no particular order.

-I think the paper could be helped by a figure or table describing the different aspects of bird aging that we do know about. It was quite a long review, and I think an illustration would help break it up.

-Line 581, you mention that mTOR is used in bird song acquisition which is very interesting. Do mTOR genes change with age in birds or is anything else known about mTOR in birds in general?

-Are their significant changes with age in physical function in birds? Could they be a model of frailty? Review focuses more on the molecular changes, but I think highlighting the physical changes with age is also important.

-Are there sex differences in aging in birds? As their sex determining genetics are different, can they be a model for sex differences?

-There are some minor typo and reference issues.

-While you mention some of the issues with using telomeres as a marker of aging, I was still surprised how much time you spent on it, as while telomeres get shorter with age, they do not predict mortality in mammals. I might make it clearer that telomeres are some the “best” we have with birds because they have been studied from a stress perspective but how it translates to biological aging is still unknown.

Author Response

In Harper and Holmes, the authors review what is known about aging in birds, and the potential use of Aves as models for aging research. Overall, I found the paper interesting, and I think it is a needed manuscript in the field. I do not have any major issues, just a couple questions/comments, listed below in no particular order.

-I think the paper could be helped by a figure or table describing the different aspects of bird aging that we do know about. It was quite a long review, and I think an illustration would help break it up.

Once again, I’ll leave this to your discretion Donna.  Personally, I don’t find summary tables/figures all that useful unless there is something particularly poignant to note since I always intend to read a paper in its entirety.

-Line 581, you mention that mTOR is used in bird song acquisition which is very interesting. Do mTOR genes change with age in birds or is anything else known about mTOR in birds in general?

 Other than birdsong, mTOR signaling has been studied in the context of ovarian aging in domesticated chickens (laying hens).  The relevance of these data to aging in natural populations of birds is limited.

-Are their significant changes with age in physical function in birds? Could they be a model of frailty? Review focuses more on the molecular changes, but I think highlighting the physical changes with age is also important.

Long-term, longitudinal studies of individual aging are in short supply in the avian aging literature so it is not possible to answer this question directly. 

-Are there sex differences in aging in birds? As their sex determining genetics are different, can they be a model for sex differences?

This question was raised by the other referee as well. The literature is limited, but there is either no difference among the sexes or a tendency for males to be longer-lived (reviewed in Austad and Fischer 2016).  We have not addressed this in the text, however, because we feel it does not add anything to the central theme of the manuscript.

-There are some minor typo and reference issues.

 We have addressed these issues in the revision.

-While you mention some of the issues with using telomeres as a marker of aging, I was still surprised how much time you spent on it, as while telomeres get shorter with age, they do not predict mortality in mammals. I might make it clearer that telomeres are some the “best” we have with birds because they have been studied from a stress perspective but how it translates to biological aging is still unknown.

We have addressed the issue of telomere biology in some more detail in the revised manuscript.  The heavy emphasis on telomere biology in the text stems from many ornithologists insisting on using it as a de facto measure of aging/longevity despite the clear limitations of this approach. 

Round 2

Reviewer 1 Report

Thank you for addressing my concerns, which I think has improved the manuscript substantially. I have a few additional comments, see below.

Line 14, typo “oxidpative”

Line 87/88, if you want to discuss models that are much longer-lived than their size would suggest, this needs to include the naked mole-rat (H. glaber)

Regarding the author reply to my comments on death hazard rate increases as indicators of biological aging and their lack in bird studies, I would want this issue to be addressed in the paper at least briefly, given that it is relevant to anyone who would want to consider birds as models of aging, and that it shows a clear need for more research to be done in this area.

Regarding the author reply to my initial comment on line 109, while it is true that molecular studies of aging in dogs have mostly followed such a cross-sectional approach between breeds of different size, a fairly large body of literature exists using actuarial data, be that from veterinary clinical databases, veterinary insurance databases, or animal cemetery records. Compare e.g. PMID 16788896, PMID 23613790, PMID 30870610, PMID 3199609, PMID 10619607.

As for the summary, I would reiterate my request that the authors put more emphasis on what they clearly believe is the potential for birds to be useful and provide new insights in future studies of aging, including aging in long-lived organisms. I would go so far as to say that the section in its present form is seriously underselling the point the authors are trying to make in this manuscript. The expanded summary could e.g. include some points regarding what is lacking and would as such be obvious targets for new research, e.g. actuarial aging studies, studies of age-related neurodegeneration and associated proteins, and studies of age-related mitochondrial dysfunction.

Author Response

Review of “New Perspectives on Avian Models for Studies of Basic Aging Processes”

Response to Reviewer 1

Thank you for addressing my concerns, which I think has improved the manuscript substantially. I have a few additional comments, see below.

Line 14, typo “oxidpative”

This has been fixed

Line 87/88, if you want to discuss models that are much longer-lived than their size would suggest, this needs to include the naked mole-rat (H. glaber)

There appears to be a disconnect that may have to do with different versions of the ms being considered.  The reference to body mass and longevity occurs at lines 59/60 on the revised version of the ms submitted on 5/5/2021.

The reviewer is correct in noting that naked mole rats are longer-lived than their size would predict, but this is true of other mammals (e.g. tree squirrels and humans) as well.  The aim of the ms is not to highlight the extreme longevity of select groups of mammals relative to their size. Bats were mentioned only for context.  Consequently, we do not feel the need to include naked mole rats or any other unusually long-lived species in this section.   

Regarding the author reply to my comments on death hazard rate increases as indicators of biological aging and their lack in bird studies, I would want this issue to be addressed in the paper at least briefly, given that it is relevant to anyone who would want to consider birds as models of aging, and that it shows a clear need for more research to be done in this area.

We respectfully disagree with the reviewer that this issue needs to be addressed directly in the body of the manuscript.  As noted, the aim of the review is to highlight recent research findings rather than rehashing the rationale for using birds as a model for aging research.

Regarding the author reply to my initial comment on line 109, while it is true that molecular studies of aging in dogs have mostly followed such a cross-sectional approach between breeds of different size, a fairly large body of literature exists using actuarial data, be that from veterinary clinical databases, veterinary insurance databases, or animal cemetery records. Compare e.g. PMID 16788896, PMID 23613790, PMID 30870610, PMID 3199609, PMID 10619607.

Again, the aim of the review is to highlight recent findings in the biology of aging in birds and not the strengths and weaknesses of various vertebrate models.  Given that non-avian (versus avian) reptiles are extremely interesting in that they clearly display negligible senescence it would be more germane to include a discussion of reptilian aging rather than mammalian aging if that was the goal of the review. 

As for the summary, I would reiterate my request that the authors put more emphasis on what they clearly believe is the potential for birds to be useful and provide new insights in future studies of aging, including aging in long-lived organisms. I would go so far as to say that the section in its present form is seriously underselling the point the authors are trying to make in this manuscript. The expanded summary could e.g. include some points regarding what is lacking and would as such be obvious targets for new research, e.g. actuarial aging studies, studies of age-related neurodegeneration and associated proteins, and studies of age-related mitochondrial dysfunction.

We and our collaborators (Austad, Ottinger, Williams, Miller, for example) have repeatedly made this point in both review articles and papers examining putative biochemical mechanisms of slowed aging in a variety of avian species.  We fail to see how reiterating these ideas is necessary given the focus of the review.

P